# Investigation of the Associations between Diet Quality and Health-Related Quality of Life in a Sample of Swedish Adolescents

**DOI:** 10.3390/nu14122489

**Published:** 2022-06-15

**Authors:** Callum Regan, Hedda Walltott, Karin Kjellenberg, Gisela Nyberg, Björg Helgadóttir

**Affiliations:** 1The Swedish School of Sport and Health Sciences (GIH), 114 86 Stockholm, Sweden; hedda.walltott@gih.se (H.W.); karin.kjellenberg@gih.se (K.K.); gisela.nyberg@gih.se (G.N.); bjorg.helgadottir@ki.se (B.H.); 2Department of Global Public Health, Karolinska Institutet, 171 77 Stockholm, Sweden; 3Division of Insurance Medicine, Department of Clinical Neuroscience, Karolinska Institutet, 171 77 Stockholm, Sweden

**Keywords:** diet quality, health-related quality of life, Swedish Healthy Eating Index for Adolescents 2015 (SHEIA15), Riksmaten Adolescents Diet Diversity Score (RADDS)

## Abstract

Most adolescents do not consume a high-quality diet, while self-reported mental health problems within this group are increasing. This study aimed to investigate the association between diet quality and health-related quality of life, and to explore the differences in diet quality and health-related quality of life between gender and parental education status. In this cross-sectional study, a detailed web-based recall method was implemented to determine dietary intake, which was analysed using the newly developed Swedish Healthy Eating Index for Adolescents 2015 (SHEIA15) and the Riksmaten Adolescents Diet Diversity Score (RADDS), to determine diet quality. The KIDSCREEN-10 questionnaire was used to measure health-related quality of life, and parental education was self-reported through questionnaires. Parental education was divided into two groups: ≤12 years or >12 years. The study included 1139 adolescents from grade 7 (13–14 years old), 51% were girls. The results showed that girls had higher scores for healthy eating and diet diversity but lower scores for health-related quality of life. A positive association was found between diet diversity and health-related quality of life (Adj R^2^ = 0.072, *p* = 0.001), between vegetable/fruit consumption and health-related quality of life (Adj R^2^ = 0.071, *p* = 0.002), and between healthy eating and diet diversity (Adj R^2^ = 0.214, *p* < 0.001). No association was found between healthy eating and health-related quality of life for all participants. The mean scores for healthy eating and diet diversity were significantly higher in the higher education parental group. In conclusion, higher diet diversity and increased fruit and vegetable consumption could be a strategy to improve health-related quality of life among adolescents. There is a need to promote better diet quality, especially in households of low parental education. In addition, there is a further need to investigate the potential benefits of improved diet quality on mental health and overall well-being.

## 1. Introduction

Adolescents across Europe often are not meeting the national dietary guidelines [1], thus, not consuming a high-quality diet. Concurrently, the number of adolescents reporting poor mental health is increasing [1]. Mental health is complex, with a variety of environmental and personal factors contributing to poor mental health status, including diet quality [2]. A high-quality diet includes: the consumption of nutrient-dense foods such as fruits, vegetables, and wholegrains, while limiting intake of red and processed meat, salt, and added sugar [3]. Existing evidence from systematic reviews has shown a positive association between diet quality and mental health in adolescents [4,5]. Additionally, a recent systematic review also found a positive association between diet quality and health-related quality of life in adolescents [2]. However, very few of the studies included in the abovementioned systematic reviews [2,4,5] have incorporated diet diversity within the scope of diet quality. Therefore, studies that specifically investigate the association between diet diversity and health-related quality of life are needed. This current study investigates this association using a newly designed and population-specific diet diversity index.

Adopting dietary guidelines is a key strategy for improving diet quality in adolescents [6]. The World Health Organization (WHO) reported an estimated number of two out of three adolescents in the European region and Canada that do not consume enough nutrient-rich foods [1]. Furthermore, results from the national Swedish survey Riksmaten Adolescents from 2016 to 2017 found that the average consumption of fruits and vegetables among Swedish adolescents was 50% less than the recommended 500 g daily amount [7]. The same report also found that participants consumed more red meat and charcuteries than recommended, and that 17% of the overall intake came from candy, cookies, and sugar-sweetened beverages [7]. Furthermore, in a population-based survey of Swedish adolescents, girls were found to consume a healthier diet than boys and 13–17-year-old adolescents were shown to adhere to unhealthier dietary patterns than 7–12-year-old adolescents [8].

To assess diet quality in adolescents, several approaches can be utilised, such as: adherence to a Mediterranean diet [9] or measuring unhealthy dietary patterns [5]. Another approach is to use dietary indices that are in accordance with dietary guidelines and recommendations [10]. The Swedish Healthy Eating Index for Adolescents (SHEIA15) and the Riksmaten Adolescents Diet Diversity Score (RADDS) were developed to capture healthy eating and diet diversity, respectively, as a measure of diet quality in Swedish adolescents [10] and were utilised in this study.

Investigating the association between diet quality and mental health in adolescents is important due to the global prevalence of the latter, accounting for 16% of disease burden amongst 10–19-year-old adolescents [11]. Globally, around 10% of children and adolescents experience mental health challenges [12], and a recent cross-national study found that 22% of adolescents (*n* = 6245) across eight European countries reported at least one mental health disorder [13]. Girls tend to report more life satisfaction complaints, for example, feeling low and nervous and are more likely to report multiple health complaints as compared with boys [14]. Furthermore, the results from the Health Behaviour in School-Aged Children (2017/2018) showed that the percentage of Swedish adolescents aged 11 to 15 years old who felt nervous or depressed had doubled since the 1980s and these health complaints were more prevalent in girls than boys [15]. Moreover, this study found that Swedish children were one of the highest groups to report these health complaints as compared with children from other European countries [15].

One way to assess mental health in adolescents is to measure health-related quality of life, which is a subset of mental health. Health-related quality of life measures the perceived physical, emotional, mental, and functional well-being of an individual [16]. It is crucial to measure domains of health in adolescents to prevent more serious long-term conditions [17]. Lower health-related quality of life has been associated with high mental health symptoms in adolescents and therefore health-related quality of life should be explored in this population [18].

The association between diet quality and health-related quality of life or other aspects of mental health in adolescents has been investigated. A previous cross-sectional analysis showed that a healthy diet, where higher scores indicated that more healthy dietary practices were being followed, was associated with higher health-related quality of life scores [19], and a past prospective cohort, measuring diet quality in a similar way, showed that improved diet quality was associated with positive mental health outcomes [20]. Additionally, a systematic review reported a dose-response relationship between health-related quality of life and diet quality, where low diet quality was associated with lower health-related quality of Life [2]. A possible explanation for the association between a low-quality diet and low health-related quality of life, or poor mental health is that diets relatively low in nutrient density can lead to nutrient deficiencies, which, in turn, are associated with mental health problems [4]. However, the possibility of reverse causality cannot be ignored [5]. Potential confounders also need to be considered when investigating the association between diet quality and health-related quality of life, such as parental education, which can have a significant influence on overall adolescent health-related quality of life [21].

Parental education is a major risk factor in developing mental health problems in adolescents [21], and can influence diet quality [22], hence, the importance to incorporate this factor when investigating the association between diet quality and health-related quality of life. Swedish adolescents with parents who have higher education have been found to have healthier dietary patterns [8], with one recent cross-sectional study finding high scores for healthy eating and diet diversity to be associated with higher education households [10]. It is proposed that parents with more educational experience are more likely to make healthier food choices for their family attributed to better nutritional knowledge [22]. 

Adequate nutrition is pivotal for growth and the increasing physiological needs of adolescents [20]. Given the relatively high prevalence of self-reported mental health problems in Swedish adolescents [14], it is important to investigate the association between diet quality and health-related quality of life in this population. It is also pivotal to explore different measures of diet quality and to examine the influence that following national dietary guidelines has on health-related quality of life. Healthy eating and diet diversity have both been proposed to measure diet quality and have been investigated separately with health-related quality of life. Thus, the aim of the current study was to investigate the association between healthy eating and diet diversity (diet quality) with health-related quality of life (a measure of mental health) in a sample of 13–14-year-old adolescents in Sweden. Additionally, potential differences in scores for healthy eating, diet diversity, and health-related quality of life between gender and parental education status were explored.

To the best of our knowledge, this is the first study to investigate the association between diet quality and health-related quality of life in Swedish adolescents. Relatively few studies have investigated this association in healthy adolescent populations [5]. Moreover, this study incorporates diet diversity into diet quality which is unique and is the first to utilise the newly developed diet quality indices SHEIA15 and RADDS that follow Swedish national dietary guidelines [10] in relation to health-related quality of life.

## 2. Materials and Methods

### 2.1. Study Design and Study Population

This study utilised a cross-sectional method. This study included a final sample size of 1139 adolescents from grade 7 (13–14-year-old girls/boys) from 34 different schools (Figure 1) within 2–3 h travel time from Stockholm.

### 2.2. Recruitment

All schools within a two–three-hour radius of the Swedish School of Sport and Health Sciences (GIH) were invited to participate in the study (*n* = 558) (Figure 1). Schools with less than 15 students in grade 7 and schools with a sports profile were excluded. The principal of each school received an invitation to participate, on 11 March 2019. The required school sample size was achieved after the first invitation; 40 schools were selected to be included, which varied in geographical location, size, and parental education status (Figure 1). One of the schools was involved in the pilot study, conducted on 3 September 2019, and six schools dropped out after initial recruitment, most likely due to time constraints (Figure 1).

Participants and guardians received an invitation letter and a consent form either through post or given directly by the teachers. Written consent was provided by both the participants and guardians before measurements could take place. The numbers of participants who declined to participate or were excluded from the study are shown in Figure 1. Only participants fluent or with a good comprehension of the Swedish language were included. The participants went to the GIH for half a day between September and December 2019, for data to be collected and for measurements, including health-related quality of life, dietary intake, body weight, and height. A week after participants visited the GIH, parental questionnaires were sent out electronically to the parents.

### 2.3. Assessment of Dietary Intake

Dietary intake was self-reported with a validated web-based method called RiksmatenFlexDiet (developed by the Swedish Food Agency, Uppsala, Sweden for Riksmaten Adolescents 2016/17); using a repeated 24 h multiple pass recall method [23], which has been validated against 24 h recall interview methods and has been shown to be reliable in recording total energy, fruit/vegetable, and wholegrain intakes [24]. Participants were given instructions and demonstrations on how to record their diets. Three dietary recording days were completed, two weekdays and one weekend day. All days were recorded retrospectively, however, day two was excluded from analyses as this was the day of the school visit and researchers at the GIH may have influenced the participants’ food choices. Thus, dietary intake was assessed using an average of two diet recording days (*n* = 1036) or just one, if only one day was completed (*n* = 97). Six participants had no dietary recording days. Participants documented their dietary intake using a food list containing 778 foods (Swedish Food Agency Food Composition Database, version Riksmaten Adolescents 2016/17), dishes, and beverages which was adapted for adolescents from Riksmaten Adolescents 2016/17 [10]. Many of these dishes and food items had corresponding photographic images which gave an indication of portion size to the participants.

### 2.4. Plausible Energy Reporters

Participants with dietary intake records for two days and with information on energy expenditure (*n* = 1008) were classified into under, plausible, or over-energy reporters from a calculated ratio of energy intake (EI) and total energy expenditure (TEE). Confidence intervals (CIs) around the mean of EI/TEE were calculated based on that reported by [7]. Those with a ratio of less than 0.5114 were classified as under-reporters and those with a ratio of more than 1.1286 were classified as over-reporters. TEE was based on prediction equations using counts per minute from accelerometer data and measured body weight (kg) of participants, as described in [25].

### 2.5. Dietary Indices to Measure Diet Quality from Dietary Intake

#### 2.5.1. Swedish Healthy Eating Index for Adolescents 2015 (SHEIA15)

To assess healthy eating, the SHEIA15 index was used which was developed by [10]. The SHEIA15 index was based on the 2015 Swedish food-based dietary guidelines, referred to as “Find your way” [26] and the Nordic nutrient recommendations [3]. Components of SHEIA15, include vegetables and fruit, fibre, fish and shellfish, red and processed meat, saturated fatty acids (SFAs), mono-unsaturated fatty acids (MUFAs), poly-unsaturated fatty acids (PUFAs), added sugar, and wholegrains [10]. Composing SHEIA15 scores involved grouping food from the subcomponents, for example, red meat together, from products and composite dishes consumed by the participants. New variables were computed to sum together total intake of that food group for each participant and was done for all components of SHEIA15. The SHEIA15 scores were calculated using a ratio of actual intake and recommended intake and adapted from [27]. This could be intake in grams or energy percent values, for instance, total intakes of SFAs (C4:0–C20:0), MUFAs (C16:1–C18:1), and PUFAs (C18:2–C21:6) in grams were converted to kcal per gram and divided by total energy intake (kcal) to acquire daily energy percent values. This was also done for added sugar. The criterion for construction of SHEIA15 components can be found in [10].

Scores were given for each SHEIA15 component and scores above one were rounded to one, for example, a participant consuming 600 g of vegetables/fruits gained a score = 1.2 (600/500, with 500 g = the recommended intake value) which would then be rounded to 1.0. Mean scores for all nine SHEIA15 components from day one and day three were added together to obtain a final SHEIA15 score. Scores ranged between 0 and 9 for each participant, with higher scores indicating better healthy eating.

#### 2.5.2. The Riksmaten Adolescents Diet Diversity Score (RADDS)

To assess diet diversity, the RADDS index was used, which was developed by [10]. The RADDS index is based on food groups from “Find your way” [26] and accounts for the participant consuming a particular food group in the dietary recording period but does not consider amounts or frequencies of food consumption. There are 17 components of RADDS, including cabbage, root vegetables, fruit, red meat, poultry, oily fish, and milk (Figure 2). Composing the RADDS scores involved grouping intakes of relevant food groups together from products and composite dishes that the participants had consumed. Composing some of the RADDS components was more complex and contained more calculations than that of SHEIA15, because the intake of many RADDS components from composite dishes could not be specified in RiksmatenFlexDiet, for example, a participant could not specify that it was wholegrain pasta within composite dishes [10]. Therefore, only consumption of specific wholegrain products could be assessed; participants did not score any points if they consumed desserts with fruits, juice, and dried fruit as these were not within the dietary guidelines. The criterion for RADDS components can be found in [10].

Participants gained a point for eating a mean intake of ≥5 g for each individual RADDS component over the 2 dietary recording days. Mean scores for all 17 RADDS components from day one and day three were added together to obtain a final RADDS score. The scores ranged between 0 and 17 for each participant, with higher scores indicating better diet diversity.

### 2.6. Assessment of Health-Related Quality of Life

Health-related quality of life was measured using the KIDSCREEN-10 (KS-10) index and was completed by participants at the GIH where they were given instructions on how to fill in the questionnaire. KS-10 is a general 10 item questionnaire shortened from the Kidscreen-52 and -27 versions aimed at children and young adults aged 8–18 years and provides one global score for health-related quality of life [28]. The KS-10 questionnaire provides a global score from items in the longer KS versions. The dimensions represented in the longer versions include physical well-being, psychological well-being, moods and emotions, self-perception, autonomy, parent relations and home life, social support and peers, school environment, social acceptance (bullying), and financial resources [28]. The KS-10 questionnaire has been found to have a 0.82 Cronbach alpha reliability score and a strong correlation with general factor scores with the longer Kidscreen-52 version [29]. Responses are measured using a five-point Likert scale, with answers ranging from “not at all” to “extremely” or from “never” to “always”. Responses were coded so that higher values were an indicator of better health-related quality of life, with scores ranging from 10 to 50 [28]. Selected questions include: Have you felt full of energy? Have you felt sad? Have you got on well at school? Have your parent(s) treated you fairly? [29]. Scores for KS-10 Items 3 and 4 were reversed, as suggested in the documentation for KS-10 [28]. The items were added up (Items 1–10) to form a scale from 10 to 50 and calculated in accordance with the Kidscreen manual [28].

### 2.7. Assessing Parental Education

Questions regarding education status were answered by the parent(s) or legal guardian(s) and their partners *(n* = 970) via parental questionnaires. Parental education status was divided by length of education experience and grouped into ≤12 years or >12 years. The parent or legal guardian with the highest level of education status in the household was used for the classification into the parental education groups. Parental education was used as a proxy for socioeconomic status (SES) and represented a SES factor.

### 2.8. Confounders

In addition to parental education, gender, body mass index (BMI), and home country were used in the regression analysis when investigating associations between SHEIA15, RADDS, and KS-10. Self-identified gender was answered by the participant and weight and height was measured through standardised procedures at the GIH for BMI calculations. BMI scores were calculated using the International Obesity Task Force age-standardised cutoffs and categorised into underweight, normal weight, overweight and obese [30]. Questions regarding home country of the adolescents were self-reported by parents/legal guardians (*n* = 1129). Parental country of birth was also reported by parents/legal guardians (*n* = 1117), and this variable was only used for descriptive purposes (Table 1).

### 2.9. Statistical Methods

The statistical analytical plan was prespecified before beginning statistical testing. The IBM Statistics SPSS 27 software was used for all data cleaning and analyses. Continuous dependent variables were all checked for normality through histogram, skewness, and kurtosis analysis. Descriptive statistics are described using the means and standard deviations for continuous variable whilst proportions of distribution are used for categorical variables (Table 1). Independent *t*-tests were used to compare the difference of age, SHEIA15, RADDS, and KS-10 between girls and boys (Table 1)Pearson’s χ2 was used to investigate distribution of proportions between categorical variables, for example, parental education and gender (Table 1). Correlation matrixes using Pearson’s R values were completed for the 3 main variables: SHEIA15, RADDS and KS-10, for all participants (Appendix A).

Multiple linear regression models were constructed to investigate the associations among SHEIA15, RADDS, and KS-10 (Table 2 and Table 3), using confounders mentioned in Section 2.8. Models 1 and 2 investigated the association between SHEIA15 and KS-10 with different confounders, whilst Models 3 and 4 investigated the association between RADDS and KS-10 with different confounders. Additionally, a multi-level mixed linear regression model was applied, with two levels: the school level and individual/student level for comparing the association between SHEIA15 and KS-10, as well as between RADDS and KS-10 (results not shown). A random intercept was modelled for each school. Assumptions for regression were checked for all regression analyses, including normality, normal PP plots of residuals, collinearity, and Durbin–Watson test. Morewover, ndependent *t*-tests were used to compare mean scores of SHEIA15, RADDS, and KS-10 between parent education groups (Table 4). A *p*-value <0.05 was used to detect statistical significance for all tests; unstandardised β coefficient 95% and CI were used for regression analyses.

## 3. Results

Participants were evenly divided by gender: girls (51%) and boys (49%), with mean ages of 13 ± 0.3 and 13 ± 0.4 years, respectively (Table 1). In addition, 71.6% of the participants had at least one parent who had >12 years of education and 85.7% of the participants stated that Sweden was the country they were born in (Table 1). A total of 1129 participants had valid dietary recalls on day one and 1040 participants for day three. The SHEIA15 (Swedish Healthy Eating Index for Adolescents 2015) scores (for healthy eating) ranged from 2.29 to 8.49, the RADDS (Riksmaten Adolescents Diet Diversity Score) scores (for diet diversity) ranged from 0 to 13, and the KS-10 scores (health-related quality of life) ranged from 18–50, with higher scores indicating better outcomes for all three variables. There were no significant Pearson’s χ2 associations between gender and the following variables: BMI (body mass index) status, parental education, country of birth, and home country (Table 1). The mean values for both SHEIA15 and RADDS scores were significantly higher in girls as compared with boys (*p* < 0.008 and *p* = 0.011, respectively), whilst the mean value for KS-10 was significantly higher in boys as compared with girls (*p* < 0.001) (Table 1).

### 3.1. Associations among SHEIA15, RADDS, and KS-10

In a multiple linear regression analysis between SHEIA15 and KS-10, no significant association was found for all participants or when stratifying for gender (Models 1 and 2, Table 2). Nonetheless a significant positive association was found between the vegetable/fruit SHEIA15 component and KS-10 for all participants when adjusting for gender, BMI, home country, and parental education (Adj R^2^ = 0.069, *p* = 0.002, unstandardised β = 1.95) and the vegetable/fruit SHEIA15 component was a significant variable in the regression model when stratifying for gender (*p* = 0.027 and *p* = 0.04, for girls and boys, respectively). Moreover, multiple regression analysis showed a significant positive association between RADDS and KS-10 for all participants when adjusting for gender, BMI, home country, and parental education (Model 4, Table 2) (Adj R^2^ = 0.072, *p* = 0.001), and RADDS was a significant variable in the regression model when stratifying for gender (*p* = 0.035 and *p* = 0.024 for girls and boys respectively) (Model 4, Table 2). Additionally, applying the multi-level mixed linear regression model, accounting for potential school clustering effect, did not change the non-significant association between SHEIA15 and KS-10 and the association between RADDS and KS-10 remained positive and significant (*p* = 0.001 for RADDS in multi-level mixed model) (results not shown). Furthermore, multiple linear regression analyses showed a positive association between SHEIA15 and RADDS for all participants when adjusting for confounders (Model 2, Table 3) (Adj R^2^ = 0.214, *p* < 0.001).

### 3.2. SHEIA15, RADDS, and KS-10 Scores in Relation to Parental Education

The mean RADDS and SHEIA15 scores for all participants as well as for girls and boys were found to be significantly higher in the >12 years parent education group as compared with the ≤12 years (*p* < 0.05) (Table 4), whereas the mean KS-10 scores were not significantly different between parental education groups (Table 4).

### 3.3. Intake and Scores of SHEIA15 and RADDS in Relation to Mean Dietary Intakes

#### 3.3.1. SHEIA15

For all participants, the mean consumption of fibre, PUFAs, vegetables/fruits, and fish/shellfish fell below their corresponding recommended values (Appendix A), whereas the mean intake of red and processed meat, SFAs, and added sugars were above recommended intakes at 500 g per week, i.e., 13.2 E% and 10.7 E%, respectively (Appendix A). The only significant difference in mean consumption of SHEAI15 components between girls and boys was that boys consumed a significantly higher amount of red and processed meat as compared with girls (98.7 ± 4 g/day as compared with 86.9 ± 3.1 g/day, *p* = 0.019). Only the consumption of MUFAs were in line with recommended values for all participants (Appendix A).

#### 3.3.2. RADDS

For all participants, the overall mean intakes for the 17 RADDS subgroups ranged between 1.29 and 197 g/d, with shellfish consumption receiving the lowest intake and milk the highest intake (results not shown). The RADDS mean scores ranged between 0.02 and 0.94 and the mean scores for RADDS components against a reference value = 1 (Figure 2). The ”milk replacements” component acquired the lowest RADDS score and ”other vegetables” component received the highest RADDS score (Figure 2). The most common vegetables contributing to the ”other vegetables” score included: cucumber, lettuce, green salad, tomato, and avocado.

### 3.4. Plausible Energy Reporters

Among the participants, 68.5% of participants were found to be plausible energy reporters, whilst 15.8% of participants were classified as under-reporters, and 15.8% of participants were classified as over-reporters (*n* = 1008). Multiple linear regression analyses were completed again for all plausible energy reporters. The main findings included: A significant association between RADDS and KS-10 for all plausible energy reporters with all confounders (*p* = 0.00). A significant association was present between SHEIA15 and RADDS (*p* = 0.00) and remained significant when confounders were added (*p* = 0.00). No significant association was present between SHEIA15 and KS-10 (*p* = 0.620).

## 4. Discussion

This study investigated associations among healthy eating, the Swedish Healthy Eating Index for Adolescents 2015 (SHEIA15), diet diversity, the Riksmaten Adolescents Diet Diversity Score (RADDS), and health-related quality of life using KS-10, in a sample of predominantly Swedish adolescents. Additionally, the comparison in scores of healthy eating, diet diversity, and health-related quality of life among parental education groups was explored. This study is unique because no other study, to the best of our knowledge, has investigated associations among these factors in this group of adolescents. Additionally, diet diversity was incorporated into diet quality which is seldom, and an association between diet diversity and health-related quality of life was found which is a novel finding.

Key findings in this study include: a significant positive association between diet diversity and health-related quality of life, a significant positive association between the fruit and vegetable component of healthy eating and health-related quality of life, a significant positive association between healthy eating and diet diversity, no significant association between healthy eating and health-related quality of life, and mean scores for healthy eating and diet diversity were significantly higher in parental education group >12 years as compared with ≤12 years.

Higher diet diversity (a component of diet quality [10]) and its association with higher health-related quality of life remained to be significant when accounting for schools, highlighting the robustness of this relationship. Although previous studies have found aspects of high quality diets to be associated with better health-related quality of life in adolescents [2,5,19], they have not found diet diversity to be positively associated with health-related quality of life. Very few studies have incorporated diet diversity into diet quality when investigating the association between diet quality and health-related quality of life, and yet this could be an important factor contributing to overall diet quality. Therefore, it is important to include measures of diet diversity in similar future investigations and to test whether the current findings could be generalizable to adolescents residing in other countries, although, it is acknowledged that other countries may need to implement their own diet quality indices tailored to specific food dietary patterns and cultural habits, for example, the Finnish Children Healthy Eating Index [31]. Since diet quality is measured in different ways, this impedes comparisons between studies investigating the relationship between diet quality and health-related quality of life; with many dietary recording methods subject to large recall errors and social desirability bias [5]. More consistent methods, for example, based on national dietary guidelines to analyse diet quality should be utilised in future studies to allow more valid comparisons to be made.

The significant positive association between healthy eating and diet diversity, as well as girls having higher scores for both, is consistent with findings from [10] and in line with findings that have shown that girls tend to make healthier food choices than boys [8,32,33]. The lower health-related quality of life in girls as compared with boys is comparable to self-reports of greater psychosomatic complaints in girls than boys [14]. However, the health-related quality of life for girls and boys on average were lower in the current sample than the recently developed aged-based norms for 11–13-year-old adolescents, where a score >42.52 was considered to be good health-related quality of life [34]. This is in line with results finding Swedish adolescents to have more mental health-related problems than other European countries [14,15]. It is difficult to know whether these differences represent a clinically meaningful difference due to the potential overestimation in classification accuracy used for the age-based norms [34]. Nonetheless, health-related quality of life (measured by mean international KS-10 t-scores, results not shown) for all participants was found to be similar as compared with a similarly aged but much larger cohort of 3283 Swedish adolescents, therefore, the current sample appears to be relatively representative of the Swedish adolescent population [29].

Higher scores for healthy eating and diet diversity in the higher education parental group is consistent with findings in [10] and comparable to findings from [8], where Swedish adolescents with parents of higher education had healthier food habits. Clearly, there are multiple points of influence by parents on household dietary intake, since parents generally act as role models, they are responsible for what is purchased in the home, and parents may influence healthier dietary habits because they themselves follow healthy dietary patterns [35].

While parental education was an important determinant of healthy eating in the current study, it may be the case that less well-educated parents are aware of dietary recommendations but find healthy foods more expensive and less accessible [36]. Therefore, advocating healthy eating practices may not only be linked to education but to overall affluence and access. Healthier food options and consuming a more varied diet are more expensive as compared with high energy-dense and nutrient-poor foods; implying that high food costs are a potential barrier for low-income households to consume a healthier diet [36]. Evidence has shown that children with parents of low educational background consume a less healthy and cheaper diet [36]. This could be an explanation for the results found in this study, however, we did not measure income, and thus, we cannot draw any definite conclusions on this matter. Nevertheless, foods that constitute a high-quality diet should be made more accessible and affordable for less affluent households.

The absence of a significant association between healthy eating and health-related quality of life may be partially due to the proportion of under-reporting or because many adolescents are not consuming a healthy diet, and even if so, this may not have been captured in the days of diet recording. However, the results from this study indicate a relationship between consuming more fruit and vegetables and higher health-related quality of life. This is comparable to results from an intervention which found that those consuming a high fruit/vegetable diet as compared with other intervention and control groups also had improvements in psychological well-being [37] and to a recent study that found those who had a higher frequency of fruit and vegetable consumption had higher health-related quality of life [33]. Furthermore, a cross-sectional analysis study in 12 different countries (*N* = 5759) found healthier food choices to be associated with better health-related quality of life, measured by KS-10 in children aged 9–11 [38]. Thus, specific food components of the diet could be important for better health-related quality of life and overall well-being, and therefore, should be investigated in future studies, rather than grouping many components into one.

The mean consumption of fruit and vegetables was more than 50% (208 g) below the recommended intake value and comparable to previous results that have shown that Swedish adolescents did not achieve the recommended intakes of fruit and vegetables [7,8,10]. Additionally, 14 diet diversity components acquired a mean score less than 0.5, which was comparable to 11 diet diversity scores (with the same diet diversity components used) receiving a score less than 0.5 in an adolescent population [10]. A relatively low proportion of under-reporters was found in this sample, indicating that these results are relatively accurate. The proportion of plausible energy reporters is higher as compared with findings in [10] and as compared with a literature review finding approximately 50% of adolescents to be plausible energy reporters from 15 different studies [39]. Based on the results from this study and previous research that showed most adolescents are not meeting recommended nutritional intakes [1], it can be concluded that awareness regarding the impact that diet quality has on aspects of mental health, needs to be promoted to help to increase adherence to dietary recommendations in this population. However, it is important to note that some of the national dietary recommendations and food guidelines may be set too high for some adolescents and not achievable for many on a daily basis, for example, 500 g of fruit/vegetables per day. Revision of the recommendations for adolescents may be worth considering to make them more realistic.

### 4.1. Limitations and Strengths

A limitation of the study is that the RiksmatenFlexDiet dietary recording method is subjective and, similar to all other self-report measures, it is prone to recall error and social desirability bias. It also calculates the average amount consumed from each of the food groups that constitute meals and not exact amounts; however, it can calculate ingredients from composite dishes [10], which is a strength. Another strength of the RiksmatenFlexDiet is that it provides photographic images and detailed descriptions of foods to aid identification of portion sizes. Additionally, mean healthy eating and diet diversity scores from dietary intake were calculated from just one or two days, and thus, perhaps did not capture food groups that were consumed less frequently, for example, fish and shellfish. Thus, increasing the days of dietary recording would be able to capture the intake of these foods, diet diversity, and healthy eating to a larger extent, and thus, a more representative picture of habitual dietary habits. This could make better use of the indices and could lead to a higher range of results on the spectrum of healthy eating and diet diversity.

There are limitations and strengths associated with using these dietary indices, since they are reliant on dietary recommendations and guidelines [3,26] within the Nordic countries including Sweden. These indices are population specific and are in line with national dietary guidelines; if they are met, they have been proposed to have important implications for health outcomes and to improve overall diet quality [3]. However, they fail to capture some foods in a potential diet. The diet diversity index, although able to capture dietary variety, does not consider the amount consumed of each component after 5 g, which is a relatively small amount. If, for example, a participant consumed a mean value of 100 g of cabbage and another 5 g of cabbage, both amounts would receive one point for the cabbage diet diversity score. Furthermore, the diet diversity index does not consider desserts with fruits, juices, and dried fruits (as they are not included in the guidelines), as well as wholegrain and egg consumption in composite dishes [10], subsequently introducing some measurement error of these food groups. It is also important to note that just because a participant scored low on either or both indices does not mean that their overall diet was unhealthy or lacking in diversity. For instance, a vegetarian/vegan would not score high in the fish and shellfish SHEIA15 component.

The use of KS-10 to measure health-related quality of life is a strength given its cross-cultural relevance and that it is convenient to use for studies with large sample sizes. KS-10 has been found to be a reliable measure for health-related quality of life in adolescents aged 8–18 years across 13 different EU countries (*N* = 22,830), including Swedish adolescents and validated against other health-related quality of life methods [29]. Moreover, using a short 10-item questionnaire may have been less burdensome as compared with the 27- or 52-item KS-10 questionnaires, and thus, using the shorter version could have provided more reliable answers regarding the participants’ health-related quality of life. Nonetheless, the KS-10 provides a global score from dimensions in the longer KS versions, and thus, significant interactions related to specific dimensions may be overlooked. A further limitation when measuring health-related quality of life is that this was only measured on one day, which may not represent participants’ overall health-related quality of life. The current analysis was of cross-sectional design, capturing only a snapshot in time, and thus, cannot infer any causations from the associations found between different exposure and outcome variables. Nevertheless, we used a relatively large sample size for this population, one sufficiently large enough to detect significance between subgroups, for example, boys and girls. Furthermore, it would have been a strength to incorporate information about parental income as a SES-F. This could have helped to explain the associations among healthy eating, diet diversity, and health-related quality of life; those with greater economy are more likely to acquire healthy diets [40].

### 4.2. Future Perspectives

Considering that most mental health problems occur in adolescence through to adulthood [20], the period of adolescence presents an ideal time for interventions to promote healthy lifestyle changes. Improving the variety and quality of foods provided to adolescents could, in turn, lead to healthier lifestyles and improvements in overall health-related quality of life, which could reduce potential future mental health problems. Thus, studies investigating the relationship between diet quality and health-related quality of life should focus on adolescent populations.

There is a need to conduct more intervention studies, such as the study carried out by Conner et al. (2017) [37] to establish causation between exposure and outcome variables and to assess if diet causes changes in health-related quality of life or vice versa. For instance, comparing differences in health-related quality of life between a high-quality diet intervention group and baseline control in a randomised controlled trial that recruits individuals already consuming unhealthy diets, could be an insightful analysis. In two recent systematic reviews only, correlational studies were identified [2,5], showing that there is a lack of intervention studies to test the causal relationship between diet and health-related quality of life.

The school lunch scheme in Sweden plays a significant role in the quality of adolescents’ diets and Sweden is among very few countries to provide school lunches free of charge [41]. The school offers hot meals, salad bars, milk, butter, and water and has been found to be more nutritious than meals provided at home [42]. School lunches may even-out diet-related inequalities associated with education, and it was found that there were not many significant differences in energy and nutrient intake in school lunches between high and low parental education groups [42]. Given this, the disparity in diet quality among different classes of parental education, may be due to food provided at home. However, it is important to bear in mind that financial constraints are likely to have an influence on diet quality at home. Therefore, a deeper investigation into the differences between diet quality at school and home in different sociodemographic areas within Sweden is recommended to identify the SES factors, for example, parental income, that have the most influence on diet quality and investigate associations among different SES factors.

In essence, future studies should incorporate multiple SES factors in analyses, for example, schools from different socioeconomic areas, schools in rural and urban areas, and parental income/wealth, since they can have major influences on diet quality [41] and on adolescents’ health-related quality of life [21]. Moreover, sample sizes should aim to represent the general population in terms of home country, country of origin, and refugees that may not speak the native language, to improve the generalisability of findings.

## 5. Conclusions

This study replicates and extends the evidence base regarding the association between diet quality and health-related quality of life in adolescents. Higher diet diversity was shown to be associated with better health-related quality of life, an insightful and unique finding in this area of research. Moreover, higher consumption of fruit and vegetables was shown to be positively associated with health-related quality of life, indicating the importance of fruit and vegetable consumption for well-being. Therefore, encouraging and promoting a varied diet and consumption of fruit and vegetables could be advocated to improve aspects of mental health in adolescents. Additionally, less healthy, and less diverse diets were found in households with parents who had experienced fewer years of formal education. However, the importance of parental income should not be overlooked, as economical constraints may be of more significance than education in obtaining healthy diets. Buying energy-dense and nutrient-poor foods and acquiring diets of lower quality have been associated with individuals who have fewer economic capabilities [40]. Thus, future studies would benefit from incorporating a variety of SES factors, especially parental income. Intervention study designs to establish causation and the incorporation of diet diversity into diet quality, when investigating the associations between diet quality and health-related quality of life, are also recommended.

## Figures and Tables

**Figure 1 nutrients-14-02489-f001:**
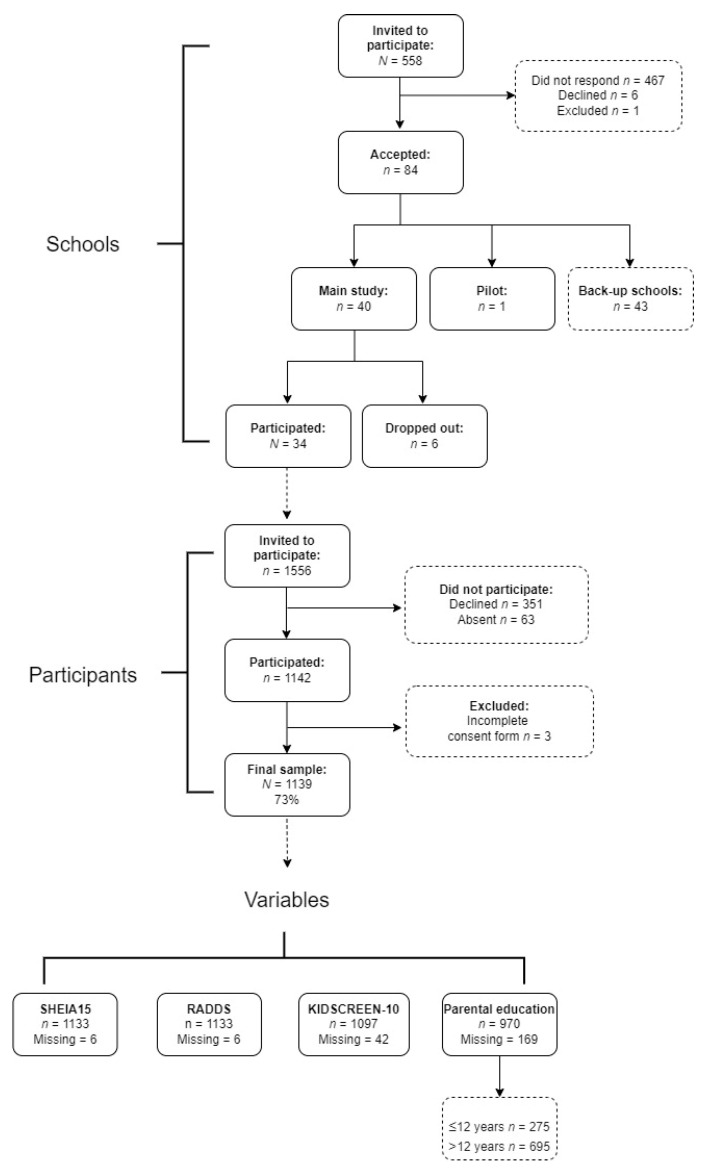
Flowchart showing recruitment of schools and participants and *N* values for variables.

**Figure 2 nutrients-14-02489-f002:**
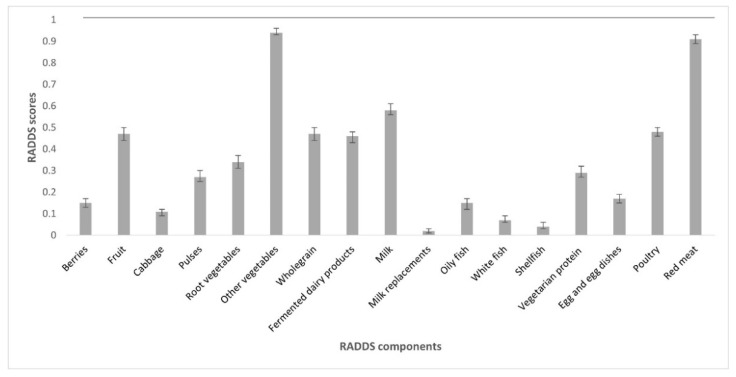
Mean scores for the Riksmaten Adolescents Diet Diversity Score (RADDS) scores for each component as compared with the RADDS reference value. Error bars = 95% confidence intervals, reference value = 1 for acquiring a RADDS score of 0.

**Table 1 nutrients-14-02489-t001:** Descriptives of participants divided by gender.

	Total *N* = 1139	Girls *n* = 580	Boys *n* = 558			
	Mean (SD)	Mean (SD)	Mean (SD)	*p*-Value ^c^	*t*-Value	DF
Age	13.4 (0.3)	13.4 (0.3)	13.4 (0.4)	0.148	−1.45	1137
SHEIA15 ^a^	5.3 (1.1)	5.4 (1.1)	5.2 (1.1)	**0.008**	2.64	1131
RADDS ^a^	5.9 (2.1)	6.1 (2.0)	5.8 (2.2)	**0.011**	2.56	1131
KS-10 ^b^	39.6 (5.4)	38.3 (5.2)	41 (5.3)	**<0.001**	−8.30	1095
	*N* (%)	*n* (%)	*n* (%)	*p*-Value ^d^		
BMI categories				0.181		
Underweight	89 (7.8)	38 (6.6)	51 (9.2)			
Normal weight	815 (71.8)	430 (74.1)	384 (69.3)			
Overweight	179 (15.8)	89 (15.3)	90 (16.2)			
Obese	52 (4.6)	23 (4.0)	29 (5.2)			
Parental education						
≤12 years	275 (28.4)	144 (29)	131 (27.8)	0.674		
>12 years	695 (71.6)	353 (71)	341 (72.2)			
Parental country of birth						
Swedish born and at least one Swedish parent	800 (71.6)	414 (72.5)	386 (70.8)	0.534		
Born outside of Sweden or both parents born outside of Sweden	317 (28.4)	457 (27.5)	159 (29.2)			
Home country						
Sweden	967 (85.7)	490 (84.9)	476 (86.4)	0.758		
EU (inc. Nordic countries)	46 (4.1)	24 (4.2)	22 (4.0)			
Outside EU	116 (10.3)	63 (10.9)	53 (9.6)			

Independent *t*-test comparing mean values of variables between gender and chi-squared frequency distribution. Abbreviations: Swedish Healthy Eating Index for Adolescents 2015 (SHEIA15), The Riksmaten Adolescents Diet Diversity Score (RADDS), KIDSCREEN-10 (KS-10), body mass index (BMI), standard deviation (SD), and degrees of freedom (DF). ^a^
*n* = 1133; ^b^
*n* = 1097; ^c^
*p*-value for independent *t*-tests; ^d^
*p*-value for Pearson’s χ2. Significant values shown in bold.

**Table 2 nutrients-14-02489-t002:** Associations between SHEIA and KS-10 and between RADDS and KS-10.

	KS-10					
	Total		Girl		Boy	
**Main Exposure:** **SHEIA**	Unstandardised β Coefficient	*p*-Value	Unstandardised β Coefficient (95% CI)	*p*-Value	Unstandardised β Coefficient (95% CI)	*p*-Value
**Model 1** SHEIA15	0.16 (−0.11, 0.44)	0.254	0.09 (−0.29, 0.47)	0.637	0.28 (−0.12, 0.68)	0.171
Gender		**<0.001**	N/A	**N/A**	**N/A**	**N/A**
Girl	REF					
Boy	2.60 (1.98, 3.22)					
**Model 2** SHEIA15	0.16 (−0.14, 0.46)	0.291	0.07 (−0.35, 0.48)	0.755	0.32 (−0.12, 0.75)	0.159
Gender		**<0.001**	N/A	N/A	N/A	N/A
Girl	REF					
Boy	2.52 (1.85, 3.19)					
BMI	−0.11 (−0.20, −0.01)	**0.024**	−0.08 (−0.22, 0.05)	0.230	−0.15 (−0.28, −0.01)	**0.034**
Home country						
Sweden	REF					
EU	−1.06 (−2.76, 0.63)	0.219	−0.74 (−3.07, 1.60)	0.535	−1.54 (−4.00, 0.92)	0.223
Outside EU	−0.79 (−1.91, 0.32)	0.163	0.17 (−1.35, 1.69)	0.822	−2.08 (−3.74, −0.42)	**0.014**
Parental education						
≤12 years	REF					
	−0.15 (−0.92, 0.62)	0.700	0.65 (−0.41, 1.71)	0.228	−1.06 (−2.18, 0.06)	0.064
**Main exposure:** **RADDS**						
**Model 3** RADDS	0.28 (0.14, 0.43)	**<0.001**	0.29 (0.08, 0.50)	**0.008**	0.28 (0.07, 0.48)	**0.008**
Gender		**<0.001**	N/A	N/A	N/A	N/A
Girl	REF					
Boy	2.72 (2.10, 3.34)					
**Model 4** RADDS	0.26 (0.11, 0.42)	**0.001**	0.25 (0.02, 0.48)	**0.035**	0.25 (0.03, 0.47)	**0.024**
Gender		**<0.001**	N/A	N/A	N/A	N/A
Girl	REF					
Boy	2.65 (1.98, 3.32)					
BMI	−0.09 (−0.19, 0.00)	0.059	−0.07 (−0.20, 0.07)	0.335	−0.14 (−0.27, −0.00)	**0.047**
Home country						
Sweden	REF					
EU	−1.00 (−2.69, 0.69)	0.244	−0.68 (−3.00, 1.64)	0.565	−1.48 (0.97, 0.99)	0.235
Outside EU	−0.70 (−1.81, 0.41)	0.219	0.19 (−1.32, 1.70)	0.803	−1.90 (−3.56, −0.24)	**0.025**
Parental education		0.529	0.47 (−0.59, 1−53)	0.384	−1.04 (−2.14, 0.06)	0.064
≤12 years	REF					
	−0.24 (−1.00, 0.52)					

Linear regression models using KS-10 as a continuous dependent variable, with SHEIA15 and RADDS as independent variables. All models adjusted for confounders and stratified by gender; significant values shown in bold. Abbreviations: Swedish Healthy Eating Index for Adolescents 2015 (SHEIA15), body mass index (BMI), Riksmaten Adolescents Diet Diversity Score (RADDS), confidence intervals (CI), reference value (REF), not applicable (N/A), and KIDSCREEN-10 (KS-10). Significant values shown in bold.

**Table 3 nutrients-14-02489-t003:** Association between SHEIA and RADDS.

	RADDS	
	Unstandardised β Coefficient	*p*-Value
**Main exposure: SHEIA**	Total	Total
**Model 1** SHEIA15	0.83 (0.73, 0.93)	**<0.001**
Gender		0.122
Girl	REF	
Boy	−0.18 (−0.40, 0.15)	
**Model 2** SHEIA15	0.80 (0.69, 0.91)	**<0.001**
Gender		0.068
Girl	REF	
Boy	−0.22 (−0.46, 0.02)	
BMI	−0.06 (−0.09, −0.02)	**0.001**
Home country		
Sweden	REF	
EU	−0.24 (−0.85, −0.37)	0.433
Outside EU	−0.37 (−0.82, 0.00)	0.073
Parental education		
≤12 years	REF	
>12 years	0.34 (0.07, 0.62)	**0.015**

Linear regression model using RADDS as a continuous dependent variable with SHEIA15 as an independent variable. All models adjusted for confounders; significant values shown in bold. Abbreviations: Swedish Healthy Eating Index for Adolescents 2015 (SHEIA15), body mass index (BMI), confidence interval (CI), reference value (REF), and The Riksmaten Adolescents Diet Diversity Score (RADDS). Significant values shown in bold.

**Table 4 nutrients-14-02489-t004:** Mean scores for RADDS, SHEIA15, and KS-10 across parental education groups.

Parental Education															
	≤12 Years, Mean (SD)			>12 Years, Mean (SD)			*p*-Value			*t*-Value			DF		
	Total	Girl	Boy	Total	Girl	Boy	Total	Girl	Boy	Total	Girl	Boy	Total	Girl	Boy
RADDS	5.5 (1.9)	5.6 (1.9)	5.4 (2.0)	6.3 (2.1)	6.5 (2.0)	6.1 (2.2)	**<0.001**	**<0.001**	**0.004**	−5.17	−4.56	−2.90	965	496	468
SHEIA15	5.0 (1.1)	5.2 (1.1)	4.9 (1.1)	5.5 (1.1)	5.5 (1.1)	5.4 (1.1)	**<0.001**	**0.001**	**<0.001**	−5.31	−3.29	−4.33	965	496	468
KS-10	9.4 (5.8)	37.7 (5.6)	41.3 (5.5)	39.6 (5.2)	38.5 (5.1)	40.8 (5.1)	0.668 ^a^	0.147	0.349	−0.45	−1.45	0.94	942	486	455

Independent *t*-tests, significant values shown in bold. Abbreviations: Riksmaten Adolescents Diet Diversity Score (RADDS), Swedish Healthy Eating Index for Adolescents 2015 (SHEIA15), KIDSCREEN-10 (KS-10), and standard deviation (SD), independent *t*-tests, ^a^ Equal variances not assumed. Significant values shown in bold.

## Data Availability

The datasets are not available for download to protect the confidentiality of the participants. The data are held at The Swedish School of Sport and Health Sciences.

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
