# Peer review of "Investigation of the Associations between Diet Quality and Health-Related Quality of Life in a Sample of Swedish Adolescents"

_nutrients, 2022, doi:10.3390/nu14122489_

Round 1

Reviewer 1 Report

The proposal about “Investigation of the Associations between Diet quality and Health-Related Quality of Life in a Sample of Swedish Adolescents” is attractive. To improve, you need the next questions:

-Abstract: it is not conventional. It is very “fragmented”. Try to elaborate it again.

-Theoretical framework: it is very updated. To be prudent, try to update some references if you find.

-Methods. This paper is based on cross sectional method. Can you justify it in depth? This study includes a size of 1139 adolescents.

-Results. In Results, the separations of text and Figures and Tables is strange (when a paper does it, the trend is to include it at the end of all the paper). I recommend you to alternate them (text, Figures and Tables in Results). Nevertheless, results are well elaborated.

-Conclusion and discussion: They are well elaborated.

Reviewer 2 Report

Investigation of the Associations between Diet quality and Health-Related Quality of Life in a Sample of Swedish Adolescents

Overall comments:  supportive of the paper but have concerns about it for the following 5 reasons.

(1)  The level of reporting of the core assessment instruments and their items, dimensions and subdomains needs to be enhanced.

(2)  There is a need to revising the reporting of the results and information in the tables along with the statistical methodology used with interval data and group comparison research.

(3)  The focus on only the global mean score for the wellness KQ-10 measure rather than the 5 dimensions. Diet is only expected to influence 2 of these 5 dimensions.

(4)  The lack of recognition that income and cost of living rather than education of parents is also a likely reason for the findings.

(5)  The lack of recommendations associated with the Swedish free lunch program to assist the diets of students in low  SES families to have more fruit and vegetables

 Comments as the paper was reviewed.

The first sentence assumes that poor diet is always associated with adolescents, this is incorrect,

Better to say: Adolescents often do not consume a high-quality diet, concurrently their self- 13 reported mental health problems are increasing

Line 90 need to put in the full name Health Related Quality of Life (HRQoL). You have so many abbreviations confusing to follow in places. The authors may know what these are, but many  reader will not.  To assist the reader’s comprehension, explain the instruments more  as you develop the paper.

The important issue is in the method section the reader needs understand the dimensions and sub-dimension that make up within each survey. Unless these are included the reader can not fully understand the study.

The term is usually SES  social economic status  so it should be SES- F if  it is social economic status – factors.  Given you are using education of parents as the status measure of SES the term status is important in this paper and should not be dropped.

It is more that education it is also income re line 106-107

“It is proposed that parents with more educational experience are more likely to make healthier food choices  for their family” . Miss the point

It also assumes that parents with more educational experience are more likely to have higher incomes and so are more likely to make and avoid healthier food choices for their family

If income is not important than the paper should be the on home education level and home diet not home SES and home diet.

See Murayama, N. (2015). Effects of socioeconomic status on nutrition in Asia and future nutrition policy studies. Journal of nutritional science and vitaminology61(Supplement), S66-S68.

Darmon, N., & Drewnowski, A. (2008). Does social class predict diet quality?. The American journal of clinical nutrition87(5), 1107-1117.

The issues raised in the well quoted  Darmon and Drewnowski paper need to be considered more.

The Swedish Healthy Eating Index for Adolescents 2015 (SHEIA15) is not well explained in this paper. The following paper did a better job of reporting it  Moraeus L, Lindroos AK, Warensjö Lemming E, Mattisson I. Diet diversity score and healthy eating index in relation to diet quality and socio-demographic factors: results from a cross-sectional national dietary survey of Swedish adolescents. Public Health Nutr. 2020 Jul;23(10):1754-1765. doi:  See their Table 5.

The KIDSCREEN-10  is not well described.  The basic information is not there in the paper: “KIDSCREEN-10 (KS-10) is derived from the KIDSCREEN-27, and provides a single index of global QoL using ten items related to physical well-being, psychological well-being, autonomy and parent relation, social support and peers, and school environment”

See how these researchers have described the KIDSCREEN-10 .

Bouwmans, C., van der Kolk, A., Oppe, M., Schawo, S., Stolk, E., van Agthoven, M., ... & van Roijen, L. (2014). Validity and responsiveness of the EQ-5D and the KIDSCREEN-10 in children with ADHD. The European Journal of Health Economics15(9), 967-977.

In terms of results the flowchart was interesting

I am a not European reviewer and so found the RADDS a rather restricted list without some meat.

Why was the Pearson's chi-squared test used (table 1) for analysis of variance?  For while  gender and in this study education are categorical (group) the data being evaluated is continuous and interval data and so an ANOVA or MANOVA by group is the method of analysis of variance. (Tabachnick, B.G., Fidell, L. S., & Ullman, J.B. (2007). Using multivariate statistics (5th ed.). Pearson.)

 In terms of gender and education: the mean, standard deviation, df and sig t or F test need to be reported in the tables.

Similarly, a correlation matrix is typically reported as it  the foundation of regression analyses and so it needs to be reported to understanding the interaction between  three  main tests  variables being investigated in this study.  

The regression analyses “p” value is reported, but the beta values and significance must also be reported. The regressing table needs reworked,  as it is the influence of diet the independent variable on wellbeing KO-10, the dependent variable. If the focus is on gender typically both a boy and a girl regression model is reported.

Table 5 is interesting but the setting out is poor and  so the columns do not align with headings, particularly the wellness KQ -10 information.  Again, an ANOVA “ t” value and df as well as the p value have to be reported.  In table 5 only one p value is reported, but what it is measuring is unclear, as there are a number of interactions occurring. Should be reporting total, then girls, and then boys as there look to be interaction effects .

Because the KQ-10 is a composite tests there is the likelihood to be some interaction with the sub-domains.  Focusing only on the global  KQ-10 scores is hiding the subdomain differences to diet. Diet is not expected to have any influence of parent relations,  social relations or peers but your study may find an influence on psychological well-being and even school environment.  This is the core of your study: does diet have an influence on psychological wellbeing? Remember the KQ-10 is made up of five subdomains (psychological well-being, autonomy and parent relation, social support and peers, and school environment) The fact is diet is no expected to change 3 of these KQ-3 subdomains  and only  one or two KQ-10 dimensions.  Use all five of the KO-10 dimensions  as your outcome measure not the Mean average global KQ-10 score. You may have a more important study if you do that, with a different finding to what you have just using the composite total.

The conclusion is sound given the findings but the lack of reference to income is an issue as educational status of parents is often a “de-facto”  measure for income. It maybe, it is the cost of living associated with  fresh fruit and vegetables that is the real issue. Given the findings the researchers could be arguing that a review may be needed with the Swedish school lunch program and increase the level of fruit and vegetables in the students' diet, particularly for students in lower SES locations. Schools may be making the lunches to a budget rather than to a healthy diet criteria which is more expensive. Different sub-populations even in the same school may need different mix of foods. A public education program could also be encouraged about health eating.

There is much about this paper that is of value, but it does need relooking at, particularly in the results reporting to do the research and the data justice.  

It needs minor revision in places but the results need more attention. Have given it minor but it is needs  more in the tables 
